# Defense-Related Gene Expression Following an Orthotospovirus Infection Is Influenced by Host Resistance in *Arachis hypogaea*

**DOI:** 10.3390/v13071303

**Published:** 2021-07-05

**Authors:** Michael A. Catto, Anita Shrestha, Mark R. Abney, Donald E. Champagne, Albert K. Culbreath, Soraya C. M. Leal-Bertioli, Brendan G. Hunt, Rajagopalbabu Srinivasan

**Affiliations:** 1Department of Entomology, University of Georgia, Athens, GA 30606, USA; mac65630@uga.edu (M.A.C.); dchampa@uga.edu (D.E.C.); 2Department of Entomology, University of Georgia, Tifton, GA 31793, USA; stha.anu@gmail.com (A.S.); mrabney@uga.edu (M.R.A.); 3Department of Plant Pathology, University of Georgia, Tifton, GA 31793, USA; spotwilt@uga.edu; 4Department of Plant Pathology, University of Georgia, Athens, GA 30602, USA; sorayab@uga.edu; 5Department of Entomology, University of Georgia, Griffin, GA 30223, USA; huntbg@uga.edu

**Keywords:** tomato spotted wilt orthotospovirus, field-resistant peanut cultivars, Sunoleic 97R, Tifguard, differential gene expression

## Abstract

Planting resistant cultivars is the most effective tactic to manage the thrips-transmitted tomato spotted wilt orthotospovirus (TSWV) in peanut plants. However, molecular mechanisms conferring resistance to TSWV in resistant cultivars are unknown. In this study, transcriptomes of TSWV-susceptible (SunOleic 97R) and field-resistant (Tifguard) peanut cultivars with and without TSWV infection were assembled and differentially expressed genes (DEGs) were compared. There were 4605 and 2579 significant DEGs in SunOleic 97R and Tifguard, respectively. Despite the lower number of DEGs in Tifguard, an increased proportion of defense-related genes were upregulated in Tifguard than in the susceptible cultivar. Examples included disease resistance (R) proteins, leucine-rich repeats, stilbene synthase, dicer, and calmodulin. Pathway analysis revealed the increased downregulation of genes associated with defense and photosynthesis in the susceptible cultivar rather than in the resistant cultivar. These results suggest that essential physiological functions were less perturbed in the resistant cultivar than in the susceptible cultivar and that the defense response following TSWV infection was more robust in the resistant cultivar than in the susceptible cultivar.

## 1. Introduction

Peanut (*Arachis hypogaea* L.) is an important crop for oil and protein production [1]. Spotted wilt disease caused by thrips-transmitted tomato spotted wilt orthotospovirus (TSWV) severely affects peanut production in southeastern United States [2,3]. In Georgia alone, spotted wilt disease causes an annual loss of over USD 10 million [4,5,6]. *Tomato spotted wilt orthotospovirus* is the species type of the genus *Orthotospovirus* in the family *Tospoviridae*. TSWV is an ambisense RNA virus that is exclusively transmitted in a persistent and propagative fashion by the peanut-colonizing thrips species in the family Thripidae [7,8,9]. Tobacco thrips, *Frankliniella fusca* [Hinds], and western flower thrips, *Frankliniella occidentalis* [Pergande], are the predominant vectors of TSWV in southeastern United States [5,10].

The use of resistant cultivars is one of the most effective tactics used for managing the incidence of TSWV and reducing the impact of spotted wilt in peanut production [2,3]. Intensive breeding programs since the 1990s in Georgia and Florida have led to the development of three generations of resistant cultivars with incremental field resistance to TSWV in each generation [3,11,12]. Under TSWV pressure, field-resistant cultivars display less severe symptoms and produce higher yields when compared with TSWV-susceptible cultivars (Figure 1) [3]. Despite the extensive usage of TSWV field-resistant cultivars over the past two decades, information on host-virus interactions at both macro and micro levels is limited, and the mechanism of resistance is unknown.

In contrast to peanut, in other crops, genes conferring resistance to TSWV have been identified and the resistance mechanism has been characterized. For instance, dominant genes *Sw-5* and *Tsw* in tomato (*Solanum lycopersicon* L.) and pepper (*Capsicum annuum* L.), respectively, have been known to confer resistance to TSWV [13,14,15,16]. Upon TSWV inoculation, these genes induced a hypersensitive reaction in TSWV-resistant tomato and pepper cultivars [16,17]. The hypersensitive reaction is characterized by localized cell death, suppressed virus replication, and a lack of systemic movement of the virus [18].

Gene(s) conferring resistance to TSWV have not been identified in field-resistant peanut cultivars, but several quantitative trait loci (QTL) believed to be involved in field resistance have been mapped [19,20,21,22,23]. Unlike in tomato and pepper, TSWV-resistant peanut cultivars do not produce a hypersensitive response upon inoculation. An earlier study demonstrated that thrips-mediated and mechanical inoculation of TSWV field-resistant and susceptible cultivars produced typical TSWV symptoms [24,25]. Those studies also documented the reduced TSWV accumulation in some resistant cultivars when compared with the susceptible cultivar. The underlying molecular basis for reduced accumulation of TSWV in field-resistant peanut cultivars remains unknown. Earlier studies have documented molecular changes induced by bacterial and fungal infections in peanut plants at the transcript levels [26,27,28]. Moreover, a repertoire of genes in peanut plants that were induced by various abiotic factors was identified [29,30,31]. Recently, affordable short-read sequencing technology is being extensively used for elucidating interactions between plant hosts and pathogens at genomic and transcriptomic levels. Studies have widely used RNA sequencing to examine the expression levels of genes following pathogen infection and to characterize genes conferring resistance to pathogens [26,27,30]. In addition, genome assemblies of the peanut sub-genomes, derived from two diploid ancestors (*Arachis duranensis* (Karpov and W. C. Gregory) and *Arachis ipaensis* (Karpov and W. C. Gregory)) [32], and two allotetraploid genomes from a United States-based cultivar [33] and a Chinese-based cultivar [34], have also become available. These genomic resources are critical to annotating the transcriptional changes in peanut cultivars following TSWV infection.

In plants, the defense against pathogens such as TSWV could be constitutive and/or induced [35,36,37]. The main objective of this study was to compare transcriptomes of TSWV-field-resistant and TSWV-susceptible peanut cultivars with and without TSWV infection. Transcriptome analysis of peanut cultivars infected with TSWV would provide insights into virus-induced changes in plant physiology and identify differentially expressed genes that encode proteins relevant to virus resistance. For this purpose, leaf tissue from greenhouse-grown TSWV-susceptible cultivar SunOleic 97R [38] and TSWV-resistant cultivar Tifguard [39] was obtained, subjected to RNA sequencing, and transcriptomes were assembled using a reference genome.

## 2. Materials and Methods

### 2.1. Maintenance of Non-Infected and TSWV-Infected Peanut Plants

Peanut cultivars SunOleic 97R and Tifguard were used for this study. Non-infected and TSWV-infected plants of each cultivar were generated as per an already established protocol [24]. Seeds from each cultivar were pre-germinated in moistened paper towels and incubated in a growth chamber at 25–30 °C, 40–50% relative humidity (RH), and L14:D10h photoperiod. The peanut seeds were then transplanted into 10cm wide pots (Hummert International, St. Louis, MO, USA) filled with a commercially available potting mix (LT5 Sunshine mix, Sun Gro Horticulture Industries, Bellevue, WA, USA). The plants were housed in thrips-proof cages (47.5 cm^3^) (Megaview Science Co., Taichung, Taiwan, CHN) from 25 to 30 °C, 80 to 90% RH, and a photoperiod of L14:D10 h in the greenhouse.

Peanut plants of the cultivar Georgia Green infected with TSWV were initially collected from the Belflower Farm, USDA, Tifton, GA, in 2009. Plants were maintained in a greenhouse under conditions as described above. TSWV-infected leaflets were enclosed in a Plexiglas^®^ cage (11.43 × 8.89 × 1.77 cm^3^) and non-viruliferous *F. fusca* were released. Following the emergence of the next generation of potentially viruliferous adults, and up to two-day-old female adults, were transferred to a 1.5 mL microcentrifuge tube (Fisher, Pittsburgh, PA, USA). Potentially viruliferous thrips (10 per plant) were subsequently released on one-week-old SunOleic 97R and Tifguard plants. Each plant with thrips was enclosed in a Mylar film (Grafix, Cleveland, PA, USA) cage with a copper mesh top. Plants were maintained in the greenhouse for three weeks. Non-infected and TSWV-infected plants of each cultivar were tested with ImmunoStrips^®^ (Agdia, Elkhart, IN, USA) to assess TSWV-infection status or lack thereof.

### 2.2. Total RNA Extraction, Library Preparation, and Sequencing

Total RNA was extracted from leaf samples of about four-week-old TSWV-infected and non-infected SunOleic 97R and Tifguard plants (four treatments). For each replicate, leaf samples were pooled from three plants and there were three replicates per treatment. Thus, a total of 12 samples were prepared (four treatments × three replicates). Total RNA was extracted using an RNeasy plant mini kit following the manufacturer’s protocol (Qiagen, Valencia, CA, USA). Subsequently, cDNA libraries were constructed at the Georgia Genomic Facility at the University of Georgia. Prior to library construction, an Agilent 2100 Bioanalyzer (Agilent Technologies, Santa Clara, CA, USA) was used to evaluate the RNA quality and concentration in the samples. Illumina sequencing libraries were constructed using TruSeq RNA sample preparation kits. Briefly, mRNA was selected, fragmented, and first-strand cDNA was synthesized using random primers and reverse transcriptase. Subsequently, Polymerase I and RNase H were used to make the second-strand cDNA. An Illumina TruSeqLT adapter was ligated to the DNA fragments, and PCR amplification was performed for a minimal number of cycles with standard Illumina primers to produce the final cDNA libraries. Twelve libraries were constructed and sequenced in two lanes (six libraries in each lane) of the Illumina HiSeq 2000 platform using v3 paired-end 100 cycle sequencing settings.

### 2.3. Transcriptome Assembly

Prior to transcriptome assembly, the read quality was checked with fastQC v0.11.8 [40]. Adapter sequences were removed from the raw reads, and the quality of the reads was determined using the default settings in Trimmomatic v0.39 [41]. Subsequently, reads from TSWV-infected and non-infected SunOleic 97R and Tifguard samples were independently aligned to a peanut reference genome [33] using a STAR v2.7.2 aligner [42] with default parameters. Trinity v2.10.0 [43] software was used to perform the reference-guided assembly using the parameters “--genome_guided_max_intron 10,000 and --min_kmer_cov 3” following the Bertioli, et al. protocol [33]. Over assembly issues were addressed with sra2genes v4 (http://arthropods.eugenes.org/EvidentialGene/other/sra2genes_testdrive/, accessed on 15 March 2021) with the default settings. The first step in this process was the removal of all duplicated sequences using fastanrdb from Exonerate [44]. The removal of potentially chimeric or misassembled transcripts was carried out using the CD-HIT-EST in the sra2genes pipeline [45]. The final filtering steps, aimed to separate transcripts that were isoforms, were accomplished via sequence clustering with a reciprocal BLAST search [46]. The post-assembly processing with sra2genes resolved the issue of read ambiguity by cross-referencing the genome for potential paralogs and homeologs [47]. The completeness of core genes in the assembly was assessed with BUSCO v4.0.6 against the Fabales Odb10 lineage [48]. The assembly pipeline provides an overview of the steps taken to process the raw read files (Appendix A).

### 2.4. Differential Expression Profiling

Following the transcriptome assembly, reads from each replicate were aligned to the assembled contigs and scaffolds using Bowtie2 v2.4.1 [49], and the relative abundance of each contig was estimated using the RSEM program [50,51]. Subsequently, multiple pairwise comparisons were carried out for differential expression analysis with DESeq2 v3.12 [52]. To assess the significance of gene expression differences, transcripts with a false discovery rate (FDR) of ≤0.05, and at least a 16-fold change (FC) in expression, were used as thresholds. The comparison of correlations was carried out using the paired.r function (https://www.rdocumentation.org/packages/psych/versions/2.1.3/topics/paired.r, accessed on 7 April 2021), which used a correlated t-test for each comparison.

### 2.5. Functional Annotation

Genes were annotated by comparison with the National Center for Biotechnology Information non-redundant (NCBI nr) database using the Blastx algorithm [46,53,54] with an E-value cutoff of 1.0 × 10^−5^ and a high-scoring segment pair (HSP) length cutoff of 33. The characterized genes were assigned Gene Ontology (GO) terms under three main categories: biological process, molecular function, and cellular component using the GO database through Blast2GO [55]. GO terms (functional annotations) were assigned to genes with an E-value of 1.0 × 10^−6^. Subsequently, the Kyoto Encyclopedia of Genes and Genomes (KEGG) [56,57] analysis using the default parameters was performed to identify the biochemical pathways in DEGs. All annotations were performed using OmicsBox v1.4.11 [58,59]. Visualization of GO terms was performed using WEGO 2.0 [60,61] and a custom R script. For gene enrichment comparisons, the enrichment ratio was calculated using the log_2_ ratio of DEGs (upregulated or downregulated) within the total/non-significant genes within the total.

## 3. Results

### 3.1. Transcriptome Assembly

The raw reads from both TSWV-infected and non-infected plants were used in the reference-guided transcriptome assembly. The total number of raw reads were 217,620,345 and 213,498,141 from SunOleic 97R and Tifguard, respectively, regardless of infection status. Filtering of the raw reads with Trimmomatic yielded 184,695,923 and 180,476,999 high quality paired-end reads from SunOleic 97R and Tifguard, respectively. All samples had >90% of the reads mapped to the allotetraploid genome (Table 1). Trinity, which has been shown to be effective for polyploid *de novo* assemblies [62], assembled these reads into 397,388 contigs with an average length of 957.745 bp. Filtering of the assembly with sra2genes reduced the assembly to 128,045 contigs with an average length of 1348.61 bp. There were 157,016 gene models for the *A. hypogaea* (peanut) genome under the NCBI GenBank assembly (GCF_003086295.2), with an average length of 1565.80 bp. The total completeness was 4070 (76%), which included 1613 (30%) single-copy and 2457 (46%) duplicated orthologs (Appendix A).

### 3.2. Quantitation of Differentially Expressed Genes

Using RNA-Seq by Expectation-Maximization (RSEM) with the built-in Bowtie-2 package, an average of 75.87% of the reads for each replicate were mapped back to the assembled contigs. Differential expression analysis was conducted on the SunOleic 97R and Tifguard cultivars separately. The fragment per kilobase of transcript per million mapped reads (FPKM) values for the normalized samples aligned to the transcriptome showed comparable density across all of the samples (Appendix A). The total number of DEGs for the SunOleic 97R cultivar was 4605 (3323 upregulated and 1282 downregulated in TSWV-infected versus non-infected samples). The total number of DEGs for the Tifguard cultivar was 2579 (2223 upregulated and 356 downregulated in TSWV-infected versus non-infected samples). Clustering of the samples through principal component analysis (PCA) showed that the samples from both cultivars were separated by TSWV infection (Appendix A). One sample, Non-inf_Tif_1, was removed at this stage due to reduced quality (Appendix A) and unexpected clustering during the principal component analysis (Appendix A). Additional PCAs were performed on the SunOleic 97R as well, by selectively removing non-infected samples to check for alterations in clustering (Appendix A). The total number of DEGs was increased with the exclusion of the reduced quality Tifguard sample, whereas the SunOleic 97R showed stability in total DEGs with the selective removal of samples (Appendix A). To check the relatedness between all of the samples per cultivar, sample-to-sample distances were computed (Appendix A). Samples of TSWV-infected SunOleic 97R showed a high number of genes (4605) that were differentially expressed compared with the samples from non-infected plants. SunOleic 97R DEG results (Figure 2 and Appendix A) revealed more genes over the log_2_ FC ≥ 4 compared with Tifguard (Figure 3 and Appendix A).

### 3.3. Functional Annotation of Genes after TSWV Infection

To determine the functional classification of genes, annotation with Gene Ontology (GO) terms was performed. For SunOleic 97R, a total of 2700 upregulated and 998 downregulated genes were annotated. There were 25,039 non-significant genes included in the analysis to check the DEGs against the background for significant enrichment. For the Tifguard cultivar, a total of 1889 upregulated and 289 downregulated genes were annotated. There were 14,306 non-significant genes included in the analysis.

Within the GO annotations, significantly enriched GO terms down to level 3 terms were chosen for presentation (Appendix A). Two lists of significantly enriched GO terms across all levels (1–6) are included in Appendix A. Across all significant GO terms in the SunOleic 97R cultivar, there were 35,783 upregulated and 14,168 downregulated DEGs, respectively. Across all significant GO terms in the Tifguard cultivar, there were 23,515 upregulated and 3236 downregulated DEGs, respectively.

To understand the biological aspects of TSWV infection, Biological Process GO terms were investigated further. There were significantly more enriched GO terms overall for SunOleic 97R than Tifguard. The largest clusters of upregulated DEGs, with the highest significant enrichment in SunOleic 97R, were found mainly within the following categories: primary metabolic processes, organic substance metabolic processes, nitrogen compound metabolic processes, metabolic processes, cellular processes, and cellular metabolic processes (Figure 4). The largest clusters of upregulated DEGs, with the highest significant enrichment in Tifguard, were found mainly within the following categories: cellular processes and cellular metabolic processes (Figure 5). Upregulated DEGs with slightly lower significance were found in the following categories: primary metabolic processes, organic substance metabolic processes, nitrogen compound metabolic processes, and metabolic processes (Figure 5).

### 3.4. Comparisons of DEGs between Cultivars

To identify the transcriptional changes in each cultivar relating to TSWV infection, the list of DEGs between SunOleic 97R and Tifguard were compared. The total number of DEGs present in both cultivars (*n* = 8544) was investigated. There were 3902 and 2630 DEGs exclusive to SunOleic 97R and Tifguard, respectively, and 23.5% (2012/8544) of the DEGs were shared between the two cultivars (Figure 6A). The DEGs in both cultivars were also checked for the independent and shared directionality of regulation. With at least a 16-fold change, the total number of DEGs expressed in the same direction between both cultivars was found to be 1987; whereas 25 DEGs were found to be regulated in different directions between the two cultivars (Figure 6B). The top significant genes from Tifguard in the DEGs comparison were identified as α-dioxygenase 1 and another as stilbene synthase 3-like (Appendix A).

The DEGs were grouped into three major categories: defense, phytohormones, and photosynthesis (Table 2). These categories were chosen based on studies referenced in Table 2. Overall, the upregulation of defense-related genes was numerically higher in the TSWV-susceptible cultivar SunOleic 97R than in the TSWV-field-resistant cultivar Tifguard. Upregulated defense-related DEGs were higher in SunOleic 97R in 45% and higher in Tifguard in 50% of the 20 examined categories. The upregulated defense-related DEGs were the same in both cultivars in 5% of the 20 examined categories. Examples of DEGs upregulated in SunOleic 97R include salicylic acid (SA), leucine-rich repeats (LRRs), lectins, MYB, P450s, serine/threonine protein kinases, and WRKY transcription factors. Examples of DEGs upregulated in Tifguard include dicer, heat-shock protein, mitogen-activated protein kinase, calmodulin, and stilbenes. Despite the numerical increase in defense-related DEGs in SunOleic 97R versus Tifguard, the proportion of defense-related DEGs in relation to the total DEGs was 38% higher in Tifguard than in SunOleic 97R. In all but one category, more defense-related DEGs were downregulated in SunOleic 97R in comparison with Tifguard.

There was increased upregulation of phytohormone-related DEGs in SunOleic 97R (*n* = 94) compared with Tifguard (*n* = 56). The opposite trend was found for photosynthesis/chloroplast-related DEGs, with the majority being downregulated in SunOleic 97R (*n* = 132) rather than in Tifguard (*n* = 42).

DEGs pertaining to defense, phytohormones, and photosynthesis in relation to TSWV infection were used to investigate the correlation of directionality and magnitude of gene regulation between the two cultivars (Figure 7). Defense-related DEGs showed a Pearson’s correlation coefficient of R^2^ = 0.69 between the two cultivars. The correlation coefficients of phytohormone- and photosynthesis-related DEGs were R^2^ = 0.82 and R^2^ = 0.62, respectively, between the two cultivars. The correlation coefficient of directionality for all DEGs was R^2^ = 0.78 (Appendix A). A correlation comparison between defense-related genes and the remaining uncategorized genes was statistically significant (*p* < 0.001). There was also statistical significance (*p* < 0.001) in the correlation comparison between the photosynthesis and uncategorized genes.

## 4. Discussion

This study examined global changes induced by TSWV infection in a TSWV-susceptible and a field-resistant cultivar at the transcript level, with the goal of gaining insights into molecular mechanisms underlying TSWV resistance. In this study, the cultivar SunOleic 97R, known for its high oleic acid level and its susceptibility to TSWV, was included [38]. In addition, Tifguard, a hybrid between the TSWV moderately field-resistant cultivar (C-99R) and the nematode-resistant but TSWV-susceptible cultivar (COAN), was included as the TSWV field-resistant cultivar [39]. Differential expression analysis between SunOleic 97R and Tifguard revealed that fewer DEGs were identified in Tifguard when compared with SunOleic 97R, suggesting that TSWV infection resulted in reduced physiological perturbance in Tifguard when compared with SunOleic 97R. In another study, it was observed that the number of TSWV-N gene copies was less in Tifguard when compared with another susceptible peanut cultivar [24]. Thus, fewer physiological and/or gene expression changes in Tifguard observed in this study could be associated with lower TSWV loads in Tifguard when compared with SunOleic 97R. Although Tifguard had fewer DEGs than SunOleic 97R, the proportion of plant defense-related genes in relation to the total differentially expressed genes was higher overall in Tifguard when compared with SunOleic 97R. Differential expression analysis identified a greater number of defense proteins in Tifguard than in SunOleic 97R in 10 over 20 categories. The upregulated DEGs in SunOleic 97R were only 1.2 times the number in Tifguard, but the number of downregulated DEGs in SunOleic 97R were four times the number in Tifguard. These results suggested that plant defense against viruses was more robust in the resistant cultivar.

Constitutive plant defense proteins, including multiple LRRs that are active against a broad range of pathogens including viruses, were higher in Tifguard than in SunOleic 97R [67,68]. Similarly, the RNAi-associated proteins that play a vital role in limiting virus replication and/or spread in plants were more upregulated in Tifguard than in SunOleic 97R [65]. More DEGs associated with heat-shock protein and the calcium signaling molecule calmodulin, known to be associated with resistance to invading pathogens such as viruses, were upregulated in Tifguard rather than in SunOleic 97R. Moreover, DEGs of the nucleocapsid N gene from tobacco, *Nicotiana glutinosa* L., that imparts resistance to several tobamoviruses including the tobacco mosaic virus (TMV), were upregulated in the resistant rather than the susceptible cultivar [90,91]. Following TMV infection, the TMV nucleocapsid N protein induced a hypersensitive reaction and restricted virus replication and movement [79]. In this study, we identified several homologs of TMV N protein upregulated in TSWV-infected Tifguard. In addition to the TMV N protein, we also observed the upregulation of several disease resistance (R) proteins that led to a hypersensitive reaction in TSWV-infected Tifguard. TSWV infection is known to induce a hypersensitive reaction in pepper and tomato [13,14,15,16]. However, in our previous studies, we observed TSWV-induced systemic infection in TSWV-resistant peanut cultivars instead of local cell death [24,25]. The upregulation of several hypersensitive reaction-inducing disease-resistant proteins suggests that these resistant proteins could play some other role in the defense against TSWV in peanut plants. One of the several known cellular functions of lectins is that they facilitate plant defense by recognizing pathogens, and they were more upregulated in the susceptible than the resistant cultivar [92]. DEGs associated with induced systemic resistance pathways, such as salicylic acid, were only present and/or upregulated in the susceptible cultivar, and not the resistant cultivar. In tomato, the disease-resistant R gene *Mi* is known to confer resistance against nematodes and potato aphids [93]. Homologs of this gene were upregulated in SunOleic 97R rather than in Tifguard. Apart from these differences in upregulated DEGs, overall, there was a significant downregulation of defense-related DEGs in the susceptible rather than the resistant cultivar, reiterating that the defense system against pathogens such as viruses is more robust and less impaired in the field-resistant cultivar, Tifguard, than in the susceptible cultivar, SunOleic 97R.

The downregulation of defense pathways leading to the production of proteins such as LRR and serine/threonine in SunOleic 97R could be attributed to, either differences in the suppression of signaling molecules or the inhibition of transcription factors associated with TSWV infection [94]. Impaired defense responses could, in turn, directly aid in sustained TSWV replication in SunOleic 97R rather than in the TSWV field-resistant cultivar, Tifguard. As well as DEGs associated with defense responses, DEGs associated with phytohormones and photosynthesis were significantly downregulated in the susceptible cultivar SunOleic 97R as opposed to the resistant cultivar Tifguard. These results suggest that, in addition to having a robust defense system, the essential functions associated with plant growth and photosynthesis were less impacted due to TSWV infection in the resistant cultivar rather than in the susceptible cultivar. These results could, in part, explain the severely altered plant phenotype, characterized by overall stunting, foliar ring spots, and yellowing, often associated with susceptible cultivars.

TSWV-resistant peanut cultivars typically exhibit field resistance to TSWV characterized by less severe TSWV-specific symptoms despite systemic infection and reduced yield losses than in the susceptible cultivars. Under field conditions, these host-virus interactions could also be influenced by abiotic factors. In plants, UV light has been shown to induce the transcription of pathogenesis-related proteins and enhance pathogen resistance [95,96]. Such a phenomenon could be true in TSWV-resistant peanut cultivars as well. In this study changes induced by TSWV infection at the transcriptional level were evaluated in greenhouse-grown SunOleic 97R and Tifguard cultivars, as it is difficult to control the TSWV inoculation process by thrips under field conditions, such as the number of thrips released per plant, thrips release timing in relation to plant age, and thrips confinement on plants. In TSWV-infected SunOleic 97R, this study identified homologs of photoreceptors, such as the blue-light photoreceptor PHR2 and phototropins that could regulate gene expression associated with plant development in response to light stimuli [97]. Greenhouse-grown plants would have had more consistent lighting compared with field-grown plants, which in turn, could differentially influence transcriptomic profiles and should be explored in future studies. Temperature is also known to substantially influence the host phenotype in peanut as well [98]. The upregulation of DEGs such as the heat-shock protein in Tifguard in this study suggests that enhanced heat tolerance could be associated with field resistance to TSWV.

TSWV-resistant peanut cultivars have been pivotal to reducing yield losses in southeastern United States, and more than 90% of the production acreage is planted with TSWV-resistant peanut cultivars. The intensive breeding programs operating in this region have primarily been responsible for the release of several such cultivars for the past two decades. However, the genomic advancements in TSWV resistance in peanut plants is relatively recent. Approximately 48 quantitative trait loci associated with TSWV resistance have recently been mapped and were used to explain up to 30% of the phenotypic variation in cultivars following TSWV infection [21]. This reiterated that TSWV resistance in peanut might not be similar to the single dominant gene-conferred TSWV resistance as in the case of solanaceous vegetables. Moreover, this study lays the foundation for providing insights into such molecular mechanisms that confer resistance to TSWV in peanut plants. This study used a single resistant cultivar, and the mechanisms operating in several cultivars should not be generalized. However, all known TSWV resistance in peanut cultivars in the United States originated from a single source–C-99R. Therefore, it is likely that there is some consistency in differential gene expression and molecular mechanisms involved in conferring TSWV resistance across TSWV-field-resistant cultivars. Nevertheless, this speculation stands to be validated.

## 5. Conclusions

TSWV-resistant cultivars have been widely used across southeastern United States and are pivotal for peanut production. Despite their prevalent usage, knowledge on host-virus interactions at a micro level and the molecular mechanisms involved in conferring resistance remain unknown. This study helps to fill that knowledge gap. Overall, based on the number of DEGs upregulated by TSWV infection in SunOleic 97R and Tifguard, this study suggests that TSWV infection potentially induces more biochemical changes in the susceptible cultivar than in the resistant cultivar. This study provides insights into how TSWV infection induces specific changes in TSWV-susceptible and field-resistant peanut cultivars at transcript levels. The information generated in this study will serve as an important resource for further investigation on the molecular factors underlying TSWV resistance in TSWV-resistant peanut cultivars.

## Figures and Tables

**Figure 1 viruses-13-01303-f001:**
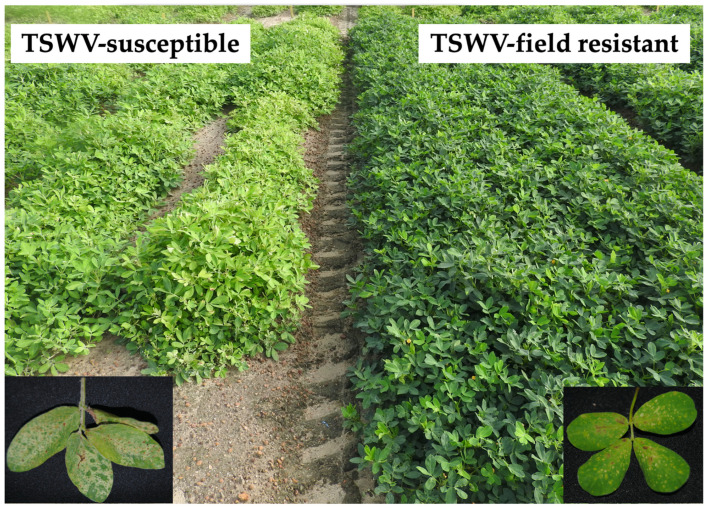
TSVW-susceptible peanut cultivar (SunOleic 97R) on the left and field-resistant peanut cultivar (Tifguard) on the right. Both cultivars were planted at the same time. TSWV infection established naturally in the selected field. As evident, there was increased yellowing and stunting in the TSWV-susceptible cultivar in comparison with the field-resistant cultivar. Foliar symptoms such as chlorotic and ring spots were also more severe in the TSWV-susceptible cultivar than in the field-resistant cultivar.

**Figure 2 viruses-13-01303-f002:**
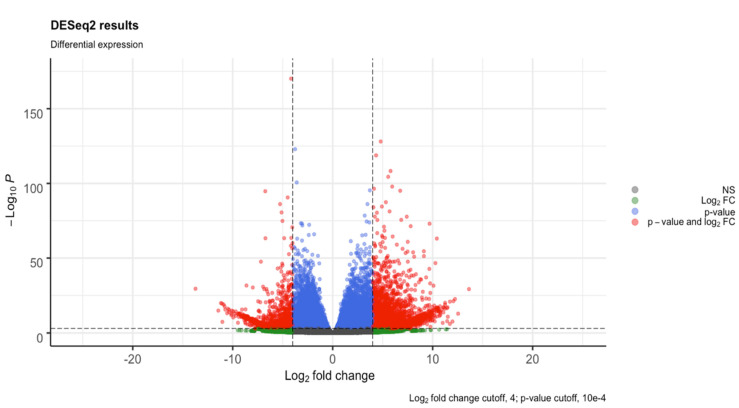
Volcano plot detailing the differential expression profiles of TSWV-infected versus non-infected samples of the cultivar SunOleic 97R. All transcripts with a *p*-value ≥ 10 × 10^−4^ and log_2_ fold-change ≥ 4 are highlighted in red (4605 genes).

**Figure 3 viruses-13-01303-f003:**
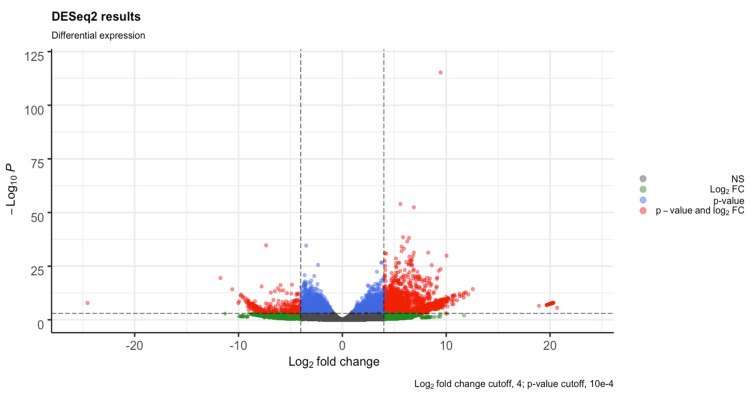
Volcano plot detailing the differential expression profiles of TSWV-infected versus non-infected samples of the cultivar Tifguard. All transcripts with a *p*-value ≥ 10 × 10^−4^ and log_2_ fold-change ≥ 4 are highlighted in red (2579 genes).

**Figure 4 viruses-13-01303-f004:**
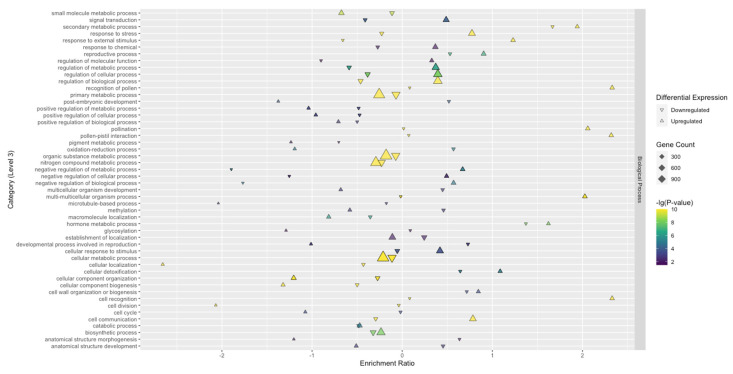
Significantly enriched level 3 GO terms categorized as Biological Processes found in the SunOleic 97R cultivar. The enrichment ratio represents the log_2_ ratio of DEGs (upregulated or downregulated) within the total/non-significant genes within the total. The upregulated genes in each category are more represented in the gene set compared with the background.

**Figure 5 viruses-13-01303-f005:**
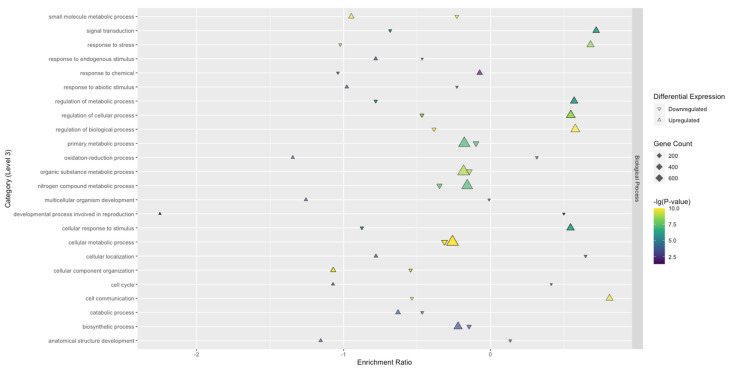
Significantly enriched level 3 GO terms categorized as Biological Processes found in the Tifguard cultivar. The enrichment ratio represents the log_2_ ratio of DEGs (upregulated or downregulated) within the total/non-significant genes within the total. The upregulated genes in each category are more represented in the gene set compared with the background.

**Figure 6 viruses-13-01303-f006:**
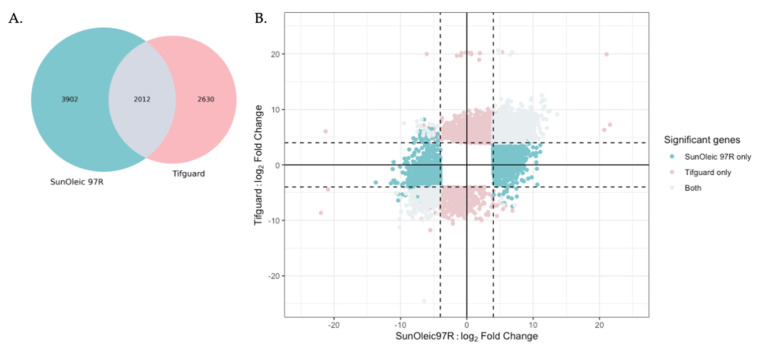
(**A**) Venn diagram of shared differentially expressed genes in SunOleic 97R (blue) with 3902 genes and Tifguard (pink) with 2630 genes in response to TSWV infection; (**B**) Log fold-change expression differences between the TSWV-infected and non-infected cultivars: SunOleic 97R (*x*-axis) and Tifguard (*y*-axis). Cutoff set for |log_2_FC| > 4 (dashed lines) and *p*-value cutoff > 0.05 in order to show DEGs per cultivar. Overlapping DEGs present in both cultivars are mostly expressed in the same direction (grey) with 2012 genes.

**Figure 7 viruses-13-01303-f007:**
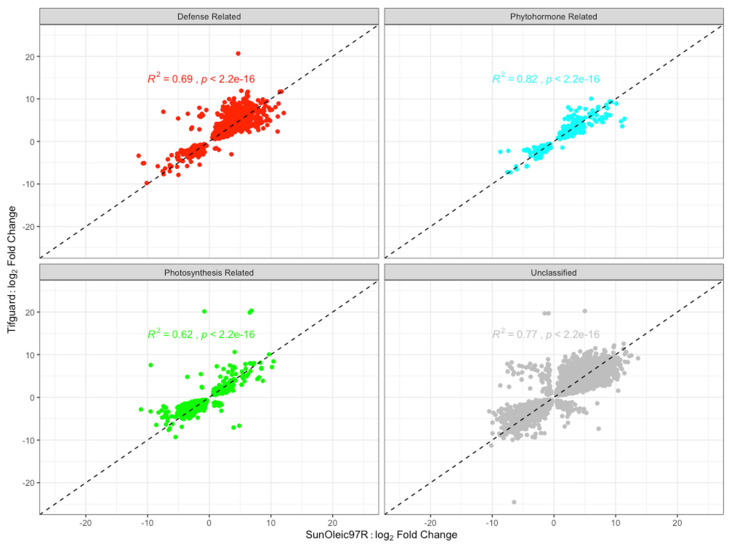
Comparison of all significant differentially expressed genes between SunOleic 97R and Tifguard in response to TSWV infection. The four categories contained the following number of genes that were shared between the two cultivars: defense (*n* = 1283), phytohormones (*n* = 299), photosynthesis (*n* = 1941), and unclassified (*n* = 9330).

**Table 1 viruses-13-01303-t001:** Percentage of reads mapped across all TSWV-infected and non-infected samples from TSWV-susceptible SunOleic 97R and TSWV-field-resistant Tifguard.

Sample ID	Number of Input Reads	Number of Uniquely Mapped Reads	Number of Multiply Mapped Reads	Total Percentage Mapped to Genome
Non-inf_Sun_5	58,031,848	33,669,316	22,372,430	96.57%
Non-inf_Sun_6	74,263,902	43,459,049	28,777,489	97.27%
Non-inf_Sun_7	73,146,084	43,046,555	28,102,807	97.27%
Non-inf_Tif_1	43,803,852	18,763,671	21,961,370	92.97%
Non-inf_Tif_4	56,785,766	33,920,234	21,250,970	97.16%
Non-inf_Tif_5	56,310,766	33,079,065	21,446,640	96.83%
Inf_Sun_4	53,110,472	29,983,015	20,227,457	94.54%
Inf_Sun_5	54,980,738	30,699,102	20,790,953	93.65%
Inf_Sun_6	55,858,802	32,665,606	20,585,822	95.33%
Inf_Tif_4	76,191,856	43,986,326	27,785,091	94.20%
Inf_Tif_5	64,818,446	37,852,846	23,687,365	94.94%
Inf_Tif_6	63,043,312	37,830,009	22,014,986	94.93%

**Table 2 viruses-13-01303-t002:** Counts of defense-, phytohormones-, and photosynthesis-related significant differentially expressed genes above the |log_2_FC| > 4 cutoff for TSWV-susceptible and field-resistant cultivars in response to TSWV infection.

Gene Description	SunOleic Upregulated	SunOleic Downregulated	Tifguard Upregulated	Tifguard Downregulated	References
**Relating to Defense**					
**Argonaute**	11	0	10	0	[63]
**MATH domain**	3	0	4	0	[64]
**Dicer**	12	0	16	0	[65]
**Heat shock protein**	3	1	6	0	[66]
**Lectin**	89	7	53	3	[67]
**Leucine zipper**	8	0	2	0	[68]
**Mitogen-activated** **protein kinase**	1	1	5	1	[69]
**MYB**	41	11	19	3	[70]
**P450**	36	16	17	4	[71]
**PAMP**	1	0	1	0	[72]
**Disease resistance (R) protein**	38	8	47	4	[73]
**WRKY transcription** **factor**	59	2	40	1	[74]
**LRR**	31	19	40	3	[75]
**Serine/threonine**	149	37	105	6	[76]
**Salicylic acid**	5	0	0	0	[77]
**Calmodulin**	28	1	37	0	[78]
**TMV resistance protein N**	76	4	86	2	[79,80]
**Stilbene synthase**	15	1	38	0	[81]
**Serine Carboxypeptidase**	17	4	9	1	[82]
**Alpha-Dioxygenase**	0	1	1	0	[83]
**Total**	623	113	536	28	NA
**Relating to Phytohormones**					
**Auxin**	10	4	2	2	[84]
**Gibberellin**	3	6	1	1	[85]
**Cytokinin**	3	0	2	0	[84]
**Abscisic acid**	10	2	3	0	[84]
**Ethylene**	24	0	7	1	[84]
**Brassinosteroid**	0	0	2	0	[86]
**Salicylic acid**	5	0	0	0	[84]
**ABC transporter**	39	8	39	1	[87]
**Total**	94	20	56	5	NA
**Relating to Photosynthesis**					
**Chloroplastic**	84	128	44	39	NA
**Protochlorophyllide**	0	3	0	1	[76]
**Photosystem**	1	6	0	6	[88]
**NADP-dependent** **malic enzyme**	1	1	1	0	[89]
**Total**	86	132 *	45	42 *	NA

* Genes counted within multiple categories. Unclassified genes were included to compare differences in genes relating to the following categories: defense, phytohormones, and photosynthesis.

## Data Availability

The raw read sequences were submitted to the NCBI SRA (https://www.ncbi.nlm.nih.gov/sra/?term= accessed on 1 February 2021). The sample submission designations are as follows: Healthy-SunOleic (SRR7196286), TSWV-infected-SunOleic (SRR7196285), Healthy-Tifguard (SRR7196284), and TSWV-infected-Tifguard (SRR7196283). The complete assembly for the transcriptome was submitted to the NCBI Transcriptome Shotgun Assembly (TSA) Database.

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
