# Peer review of "Defense-Related Gene Expression Following an Orthotospovirus Infection Is Influenced by Host Resistance in Arachis hypogaea"

_viruses, 2021, doi:10.3390/v13071303_

Round 1
Reviewer 1 Report
The Ms entitled “Defense-related gene expression following an orthotospovirus infection is influenced by host resistance in Arachis hypogaea” by Catto et al. provide some important information for further investigation on host factors underlying TSWV resistance in TSWV-resistant peanut cultivars. I think it merits to be published after some revision.
There follow a few major questions and suggested modification.
1) Symptoms and viral titer in TSWV-susceptible (SunOleic 97R) and field-resistant (Tifguard) peanut cultivars with and without TSWV infection should be clearly indicated in Figure 1. This is essential for better understanding the TSWV resistance in TSWV-resistant peanut cultivars.
2) The validation of RNA-Seq data should be further verified. My suggestion is that the authors provide RT-qPCR analysis for some DEGS that are randomly selected.
Other comments.
1) Lines 186-187. This sentence is confusing. Please clearly state that the samples from SunOleic 97R or Tifguard are with or without TSWV infection.
2) Line 270: “ ~30% (2,012) of the DEGs were”, please explain how to get the ratio 30%?
3) Lines 295-296: “with the majority being downregulated in SunOleic 97R (n = 20) than in Tifguard (n = 5)”, whether the numbers for the downregulated DEGs were correctly showed?
4) Line 34: Change tomato spotted wilt orthotospovirus to tomato spotted wilt orthotospovirus (TSWV) .
5) lines 36-37: Change Tomato spotted wilt virus to TSWV.
6) Lines 37-38: Change Tomato spotted wilt virus (TSWV) to TSWV.
Author Response
Please see file attached

Reviewer 2 Report
The authors clearly demonstrate that two cultivars of peanut (TSWV-resistant and TSWV-susceptible) infected by TSWV differ in expression of defense-related genes (and the other genes as well). As the mechanisms of resistance to TSWV infection in peanut plants has not been uncovered yet this work certainly sheds light on the possible factors involved and it is therefore valuable and necessary starting point for further studies, suitable for publication after following minor corrections:
Methods:
- pg 3, row 117: the plants without symptoms were not tested for TSWV-infection and were expected to be negative as TSWV inoculation produced typical symptoms in both susceptible and resistant plants in previous study. Could the authors address possible bias caused by this approach in the Discussion?
-pg 5, row 207 (wrong sum or numbers of particular DEGs): 4 606 is listed as sum of 3323 and 1282 DEGs, however 3323+1282 = 4605. Particular wrong value(s) should be corrected also in other places of the document they appear in (e.g. pg 6, row 220, Fig. 2, etc.)
-pg5, row 209 (wrong sum or numbers of particular DEGs): 2581 is listed as sum of 2333 and 357, however 2333 + 357 = 2690. Particular wrong value(s) should be corrected also in other places of the document they appear in.
Discussion:
- pg 12, row 322-326: "Planting TSWV... have not been examined." These sentences can be omitted as they are in introduction and in Conclusion, too.
- pg 14, row 398: "...identified homologs of photoreceptors that could regulate gene expression..." - It is not clear, which one the authors mean. P450 or some other? If P450 then the study found more of these upregulated in susceptible cultivar than in field-resistant...
- pg 14, rows 401-402: "Such photoreceptors ... could be more induced in field environment..." - to me it seems more logical that these receptors can be less induced on the field, as in the greenhouse the light is shining more constantly and regularly (in the nature the light flux is often decreased by clouds).
- Many viruses are able to inhibit defense pathway in the cells, the study found downregulation of LRR proteins (which are part of Pathogen related receptors not only in plants) and serine/threonine kinases (the key enzymes of activation cascades) in the susceptible form. This downregulation can be the results of interaction/inhibition of some messengers of defense pathway or even some transcription factors by the virus. The observed great difference in the defense-related genes expression could be thus result of just a small interaction of viral protein with some cellular one (as inhibition/hampering of defense-pathway will facilitate virus multiplication and spreading and, therefore, the other influencing of cell metabolism). Can the authors add a small paragraph to the Discussion, if such a phenomenon can explain the results observed?
Round 2
Reviewer 1 Report
This revised manuscript has addressed most of previous comments and been greatly improved. So, I think this manuscript is suitable for acceptance. However, some points below are required to be addressed appropriately.
Minor points
1. Line 315. In figure 6A legend, please clarify that the differentially expressed genes in SunOleic 97R or Tifguard were related to TSWV infection.
2. Lines 319-320. Please clarify that the differentially expressed genes in each cultivar were related to TSWV infection in table 2.
